# Fair Clustering for Data Summarization:
# Improved Approximation Algorithms and Complexity Insights

## Abstract

Data summarization tasks are often modeled as $k$-clustering problems, where the goal is to choose $k$ data points, called cluster centers, that best represent the dataset by minimizing a clustering objective. A popular objective is to minimize the maximum distance between any data point and its nearest center, which is formalized as the $k$-center problem. While in some applications all data points can be chosen as centers, in the general setting, centers must be chosen from a predefined subset of points, referred as facilities or suppliers; this is known as the $k$-supplier problem. In this work, we focus on *fair* data summarization modeled as the *fair $k$-supplier problem*, where data consists of several groups, and a minimum number of centers must be selected from each group while minimizing the $k$-supplier objective. The groups can be disjoint or overlapping, leading to two distinct problem variants each with different computational complexity.

We present 3-approximation algorithms for both variants, improving the previously known factor of 5. For disjoint groups, our algorithm runs in polynomial time, while for overlapping groups, we present a fixed-parameter tractable algorithm, where the exponential runtime depends only on the number of groups and centers. We show that these approximation factors match the theoretical lower bounds, assuming standard complexity theory conjectures. Finally, using an (anonymous) open-source implementation, we demonstrate the scalability of our algorithms on large synthetic datasets and assess the price of fairness on real-world data, comparing solution quality with and without fairness constraints.

## CCS Concepts

• **Theory of computation** → *Approximation algorithms analysis*; **Facility location and clustering**; **Fixed parameter tractability**.

## Keywords

Algorithmic fairness, Fair clustering, Responsible computing

## 1 Introduction

Data summarization is a fundamental problem for extracting insights from web data or other sources. Algorithmic fairness in data summarization is essential to ensure that the insights derived from the data are unbiased and accurately represent diverse groups. Consider, for example, a web image search for the term "CEO." An algorithmically-fair result should display a small subset of images of CEOs that accurately represent the population demographics. The summarization task can be modeled as an instance of the $k$-center problem ($k$-CENTER), where images are data points and distances between images represent their dissimilarity. We need to find a subset of $k$ data points—called cluster centers—that minimize the maximum distance from the data points to their closest center. These chosen cluster centers are then displayed as search results.

A case of algorithmic bias is well documented when for the search query "CEO", Google Images returned a much higher proportion of male CEOs compared to the real-world ratio [20]. To address such bias, Kleindessner et al. [21] introduced the *fair $k$-center problem*, where constraints are imposed to ensure that a minimum number of cluster centers of each demographic group are chosen. For example, if 70% of CEOs in the real-world are male, then for the search query "CEO" that returns ten images, about three should feature images of female CEOs. Additionally, it is possible that some images are of poor quality or contain inappropriate content and must be excluded from the search results. This consideration leads to the *fair $k$-supplier problem* (FAIR-$k$-SUP), where the cluster centers must be chosen from a specific subset of data points—called facilities or suppliers—while ensuring fair representation across groups and minimizing the maximum distance from the data points to their closest cluster center [6].

Much of the literature on fair clustering considers the demographic groups to be disjoint. However, this assumption is not realistic in modeling the real-world, where individuals belong to multiple groups, such as being non-binary, from minority ethnic groups, and/or economically disadvantaged, thus, forming *intersecting* demographic groups. Ignoring group intersections often overlook crucial nuances introduced by these intricacies and research has shown that intersecting groups often face greater algorithmic discrimination; for example, algorithms were less accurate for black women than for either black people or women individually [19]. To address intersectionality in clustering problems, Thejaswi et al. [24, 25] introduced fair clustering problems, where demographic groups may overlap, and a minimum number of cluster centers must be chosen from each group while minimizing a clustering objective, either $k$-median or $k$-means.[1]

Thejaswi et al. [24, 25] highlight that group intersectionality increases the computational complexity of fair clustering significantly. They prove that the problem is inapproximable to any multiplicative factor in polynomial time and show inapproximability even in special cases, such as when each group has exactly two facilities, when the underlying metric is a tree, and even when allowed to select $f(k)$ cluster centers (for any computable function $f$) when asked for $k$ cluster centers. On a positive note, for intersecting facility groups, they presented fixed parameter tractable algorithms (FPT),[2] yielding $\approx (1 + \frac{2}{e})$-approximation for FAIR-$k$-MEDIAN and $\approx (1 + \frac{8}{e})$-approximation for FAIR-$k$-MEANS. Although [24, 25]

---

[1]Thejaswi et al. refers to clustering problems with intersecting facility groups as *diversity-aware clustering*, as they study the problem in the context of improving diversity in clustering.

[2]A problem $\Pi$ is fixed-parameter tractable (FPT) with respect to a parameter $k$ if, for every instance $(X, k) \in \Pi$, there exists an algorithm with runtime $f(k) \cdot \text{poly}(|X|, k)$, where $f$ depends only on $k$ and $\text{poly}(|X|, k)$ is a polynomial function. The function $f(k)$ is necessarily super-polynomial for NP-hard problems (assuming $P \neq NP$), but allows efficient runtimes for small $k$, even when the input size is large. An FPT algorithm or parameterized algorithm with respect to parameter $k$ is succinctly denoted as FPT($k$).

Table 1: A summary of algorithmic results. Here, $|U| = n$ represents the number of data points, $t$ is the number of groups, and $k$ is the number of cluster centers. In Fair-$k$-Sup-$\varnothing$, the groups are disjoint, while Fair-$k$-Sup allows for intersecting groups.

| | Known results | | Our results | |
|---|---|---|---|---|
| Problem | Apx ratio | Time complexity | Apx ratio | Time complexity |
| Fair-$k$-Sup-$\varnothing$ | 5 | $O(kn^2 + k^2\sqrt{k})$ [6] | 3 | $O((kn + k^2\sqrt{k})\log n \log k)$ [Theorem 3.1] |
| Fair-$k$-Sup | 5 | $O(2^{tk}t(kn + k^2\sqrt{k}))$ [6, 24] | 3 | $O(2^{tk}k(kn + k^2\sqrt{k})\log n \log k)$ [Theorem 3.2] |

focus on complexity and algorithmic results for $k$-median and $k$-means objectives, these results can be directly extended to the fair $k$-supplier problem with intersecting groups.[3]

**Our contributions.** Our work focuses on the fair $k$-supplier problem, which is (informally) defined as follows. We are given a set of data points in a metric space that are grouped into (possibly intersecting) sets of clients and facilities. In addition, we are given a collection of groups (possibly intersecting) over the facilities, such as demographic groups defined by a set of protected attributes. Furthermore, we are given a requirement vector specifying minimum number of facilities to be chosen from each group, expressing the notion of fairness in the fair $k$-supplier problem. Finally, as it is common in clustering problems, we consider that the desired number of cluster centers, $k$ is given. The objective is to select a $k$-sized subset of facilities, which satisfies the group requirements while minimizing the maximum distance between any client to its nearest cluster center. The problem has two variants based on whether the facility groups are disjoint or intersecting.

Our main contributions are to present improved and tight approximation algorithms for both variants of the fair $k$-supplier problem, significantly advancing the state-of-the-art on the approximability. Our algorithmic results are summarized in Table 1. More formally, our contributions are as follows:

- We present a near-linear time 3-approximation algorithm for the fair $k$-supplier problem with disjoint groups.

- For the general variant with intersecting groups, we present a fixed-parameter tractable 3-approximation algorithm with runtime FPT$(k + t)$, where $k$ is the number of cluster centers and $t$ is the number of groups.[4]

- Under standard complexity theory assumptions, we show that the approximation factors match the lower-bound of achievable approximation ratios for both problem variants.

- Using an open source implementation,[5] we validate our scalability claims on large synthetic data and real-world data with modest sizes. We assess the price of fairness by comparing the clustering objective values with and without fairness constraints.

- For the fair $k$-supplier problem with intersecting facilities, our algorithm is the first with theoretical guarantees on the approximation ratio that scales to instances with modest sizes, while the earlier algorithms with theoretical guarantees struggled to scale in practice.

**Our techniques.** Here, we give a brief overview of our approximation algorithms.

For the fair $k$-supplier problem with disjoint groups, our algorithm works in two phases. In the first phase, we select a set of $k$ clients, called *good client set* $C'$, ensuring every client is at a distance 2·OPT from $C'$, where OPT is the optimal cost. To obtain such a good client set $C'$, we start by initializing $C'$ with an arbitrary client and iteratively pick a farthest client from $C'$ and add it to $C'$, for $k - 1$ times. Since $C'$ are clients, in the second phase, we recover a feasible set of facilities $S$ using $C'$, that meets the fairness requirements. This step incurs an additional OPT-factor, leading to an overall approximation factor of 3. More precisely, we guess the optimal cost OPT and construct a bipartite graph $H$ between $C'$ and the groups, adding an edge between a client $c \in C'$ to a group $G$ if there exists a facility in group $G$ within distance OPT from $c$. We show that, using $C'$ and any maximum matching in $H$, we can obtain a 3-approximate solution satisfying the fairness constraints. Finally, we can guess OPT efficiently, as there are at most $k \cdot n$ distinct distances between $C'$ and facilities.

For the fair $k$-supplier problem with intersecting groups, we build on the ideas of Thejaswi et al. [24]. We reduce an instance of the fair $k$-supplier problem with intersecting groups to many instances of the fair $k$-supplier problem with disjoint groups. The guarantee of our reduction is that there is at least one instance of fair $k$-supplier with disjoint groups whose optimal cost is the same as the original instance of fair $k$-supplier with intersecting groups. Hence, we use the above described near-linear time 3-approximation on every instance of fair $k$-supplier with disjoint groups and return the solution with minimum cost.

**Roadmap.** The remainder of the paper is organized as follows. We formally define the fair $k$-supplier problem in Section 2 and we present our approximation algorithms, including overview, proof sketches, and tight examples in Section 3. In Section 4 we present the experimental evaluation of our algorithms, showing their scalability compared to baselines. In Section 5 we discuss the related work to the problem we study. Finally, Section 6 offers a short conclusion, limitations and directions of future work.

---

[3]For algorithmic results, the techniques of [24] yield a $c$-approximation algorithm for fair $k$-supplier with intersecting groups in time FPT$(t, k)$, when given a polynomial time subroutine for $c$-approximation for fair $k$-supplier with disjoint groups.
[4]Fixed-parameter tractable in terms of $k$ and $t$ is denoted as FPT$(k + t)$ and its running time is of the form $f(k, t) \cdot \text{poly}(n, k, t)$, where $f(k, t)$ can be super-polynomial.
[5]https://anonymous.4open.science/r/fair-k-supplier-source-C60A

## 2 Problem definition

Before we present our approximation algorithms, let us formally define the fair $k$-supplier problem.

**Definition 2.1 (The fair $k$-supplier problem).** *An instance of a fair $k$-supplier is defined on a metric space $(U, d)$ with distance function $d : U \times U \to \mathbb{R}_{\geq 0}$, a set of clients $C \subseteq U$, a set of suppliers (or facilities) $F \subseteq U$, an integer $t > 1$, a collection $\mathbb{G} = \{G_1, \ldots, G_t\}$ subsets of suppliers $G_i \subseteq F$ satisfying $\bigcup_{i \in [t]} G_i = F$, an integer $k > 0$, a vectors of requirements $\vec{\alpha} = \{\alpha_1, \ldots, \alpha_t\}$, where $\alpha_i \geq 0$ corresponds to group $G_i$. A subset of suppliers $S \subseteq F$ is a feasible solution for the instance if $|S| \leq k$ and $\alpha_i \leq |S \cap G_i|$ for all $i \in [t]$, i.e., at least $\alpha_i$ clients from group $G_i$ should be present in solution $S$. The clustering cost of solution $S$ is $\max_{c \in C} d(c, S)$. The goal of the fair $k$-supplier problem is to find a feasible solution that minimizes the clustering cost.*

When the facility groups in $\mathbb{G}$ are disjoint, we denote the problem as Fair-$k$-Sup-$\varnothing$. On the other hand, for the general case, when the groups can intersect, we denote the problem as Fair-$k$-Sup.

**Remark 2.2.** *For Fair-$k$-Sup-$\varnothing$, note that $\sum_{i \in [t]} \alpha_i \leq k$, otherwise the instance is infeasible. In fact, without loss of generality, we assume that $\sum_{i \in [t]} \alpha_i = k$. This is because if $\sum_{i \in [t]} \alpha_i < k$, then we can create a new (super) group $G_0 = F$ with requirement $\alpha_0 = k - \sum_{i \in [t]} \alpha_i$. Note that, we now have $\sum_{i=0}^{t} \alpha_i = k$, and furthermore, the cost of every solution in the original instance is same as its cost in the new instance and vice-versa.[6]*

We assume that $U = C \cup F$ and we use $|U| = n$ in the analysis of time complexity. Note that $|C| = n_c \leq |U| = n$ and $|F| = n_f \leq |U| = n$, that is, both the number of clients and facilities are upper bounded by $n$.

## 3 Approximation algorithms

In this section, we present a polynomial-time 3-approximation algorithm for Fair-$k$-Sup-$\varnothing$ and prove that the approximation ratio is tight unless P = NP. Next, we extend our approach to provide a 3-approximation algorithm for Fair-$k$-Sup in FPT$(k + t)$ time and show that the approximation factor is tight assuming W[2] $\neq$ FPT. Due to space constraints, we provide only proof sketches in this section, with detailed proofs deferred to Appendix B.

**Theorem 3.1.** *There is a 3-approximation algorithm for the problem Fair-$k$-Sup-$\varnothing$ with runtime $O((kn + k^2\sqrt{k}) \log n \log k)$. Furthermore, assuming P $\neq$ NP, no polynomial-time algorithm achieves $(3 - \epsilon)$-approximation for Fair-$k$-Sup-$\varnothing$, for any $\epsilon > 0$.*

**Algorithm overview and comparison with previous work.** Let us recall the 5-approximation algorithm of Chen et al. [6], which is based on the techniques introduced by Jones et al. [18]. They first solve $k$-Supplier without fairness constraints. Towards this objective, they find a subset $F'$ of facilities that is 3-good — that is, every client is at a distance 3 times the optimal cost from a facility in $F'$. They show that selecting the farthest clients iteratively for $k$ steps and choosing the closest facility for each selected client,

gives a 3-good facility set $F'$. This approach is based on the idea of Hochbaum and Shmoys [14].

However, the set $F'$ may be an infeasible solution as it may not satisfy the fairness constraints. To satisfy fairness constraints, Chen et al. [6] build "test-swaps" in order to swap a subset of $F'$, and use a maximal-matching framework to identify a "fair-swap" (a swap, which, if performed, will satisfy the fairness constraints). To accomplish this goal, they identify a subset of suitable facilities from $F'$ to replace. "Suitable" has a twofold interpretation: first it aims to minimize the number of facilities to replace, since each substitution may increase the objective function value; second, the cost of each substitution should not be excessively high, so every "suitable" facility should be relatively easy to substitute with a nearby facility while also satisfying fairness constraints. They construct fair-swaps using a matching framework that introduces an additional factor 2 in the approximation, leading to a 5-approximation in polynomial time.

In contrast, our algorithm adopts a simpler approach (see Algorithm 1). Rather than finding a 3-good facility set in the first phase, we find a 2-good client set $C'$, that is, every client is within distance twice the optimal cost from $C'$. This has two advantages — first, instead of losing factor 3 by finding a 3-good facility set, we lose only factor 2. Second, we can find a feasible solution from $C'$ by losing only an additional factor in the approximation, rather than losing factor 2, as in [6]. This is obtained in the second phase using a matching argument.

Now, we present the algorithm and give a sketch of its correctness.

**Proof sketch of Theorem 3.1.** Our pseudocode, described in Algorithm 1, takes an instance $I = (C, F, \mathbb{G} = \{G_1, \ldots, G_t\}, k, \vec{\alpha})$ of problem Fair-$k$-Sup-$\varnothing$. Fix an optimal clustering $C^* = \{C_1^*, \ldots, C_k^*\}$ for $I$ corresponding to the solution $F^* = \{f_1^*, \ldots, f_k^*\}$, and let OPT denote the optimal cost of $F^*$. On a high level, our algorithm works in two phases. In the first phase, we find a set $C' \subseteq C$ of $k$ clients, called *good client set*, such that $d(c, C') \leq 2 \cdot \text{OPT}$, for every $c \in C$. Note that $C'$ is not a feasible solution to our problem as it is a set of clients and not a set of facilities. Hence, in the second phase, we recover a feasible solution using $C'$, which incurs an additional factor of OPT in the approximation, yielding an overall approximation factor of 3.

In more detail, we construct the good client set $C'$ by recursively picking a farthest client (breaking ties arbitrary) from $C'$ and adding it to $C'$ for $k$ iterations (see the for loop at line 3). Let $C' = (c_1, \ldots, c_k)$, where $c_i$ was picked in iteration $i \in [k]$, and let $C_i' = (c_1, \cdots, c_i)$. We claim that $d(c, C') \leq 2 \cdot \text{OPT}$ for $c \in C$. Towards this, we say that a cluster $C_i^* \in C^*$ is *hit* by $C'$ if $C_i^* \cap C' \neq \emptyset$. Let $\hat{C}_i$, for $i \in [k]$, denote the optimal clusters hit by $C_i' = (c_1, \ldots, c_i)$. If every cluster in $C^*$ is hit by $C'$, then $d(c, C') \leq 2 \cdot \text{OPT}$, as desired. Now, assume this is not the case, and hence $C'$ hits some optimal cluster at least twice, as $|C'| = k$. Then, note that as soon as $C_i'$ hits an optimal cluster twice, for some $i \in [k]$, the present set $C_i'$ is a 2-good client set. To see this, let $\ell^* \in [k]$ be the first index such that $\hat{C}_{\ell^*+1} = \hat{C}_{\ell^*}$, and let $C_i^* \in C^*$ be the cluster hit by $C_{\ell^*+1}'$ twice — by $c_{\ell^*+1}$ and by some $c_j \in C_{\ell^*}'$. Then, for any $c \in C$, we have $d(c, C_{\ell^*}') \leq d(c_{\ell^*+1}, C_{\ell^*}') \leq d(c_{\ell^*+1}, c_j) \leq d(c_{\ell^*+1}, f_i^*) + d(f_i^*, c_j) \leq 2 \cdot \text{OPT}$, since $c_{\ell^*+1}$ was the farthest client from $C_{\ell^*}'$.

---

[6]This may break the metric property of the space but our algorithms are robust to such modifications.

However, as mentioned before, $C'$ is not a valid solution. The goal of the algorithm now is to obtain a feasible set $S$ (satisfying the group constraints) using $C'$. For ease of exposition, assume that every group $G_j \in \mathbb{G}$ has a requirement $\alpha_j = 1$. Suppose we know $\ell^*$ (it can be shown that a simple binary search on $[k]$ is sufficient for recovering $\ell^*$), then consider the good client set $C'_{\ell^*} = (c_1, \dots, c_{\ell^*})$. We obtain a feasible solution using $C'_{\ell^*}$ as follows. Let $\lambda^*$ be the maximum distance $d(c_i, F^*)$ for $c_i \in C'_{\ell^*}$. We also assume that $\lambda^*$ is known (later, we show this can be obtained using binary search on a set of size $kn$). Using $C'_{\ell^*}$ and $\lambda^*$, we create a bipartite graph $H = (C'_{\ell^*} \cup \mathbb{G}, E)$ (see Lines 12–14 in Algorithm 1) where we add an edge $(c_i, G_j)$, for $c_i \in C'_{\ell^*}$ and $G_j \in \mathbb{G}$, to $E$ if there is a facility $f \in G_j$ such that $d(c_i, f) \leq \lambda^*$. Next, we find a maximum matching $M$ in $H$ on $C'_{\ell^*}$ (Line 15). A key observation is that such a matching exists in $H$ since for every $c_i \in C'_{\ell^*}$ there is a unique facility in $F^*$ in a unique group in $\mathbb{G}$ at a distance at most $\lambda^*$ from $c_i$. This is based on the fact that $C'_{\ell^*}$ hits distinct clusters in $C^*$. Once we have matching $M$ on $C'_{\ell^*}$, then for every edge $(c_i, G_j) \in M$, we pick an arbitrary facility from $G_j$ at a distance at most $\lambda^*$ from $c_i \in C'_{\ell^*}$. Again, such a facility exists due to the construction of $H$. Let $S$ be the set of picked facilities. Then, note that, for any $c \in C$, we have $d(c, S) \leq d(c, c_i) + d(c_i, S) \leq 2 \cdot \text{OPT} + \lambda^* \leq 3 \cdot \text{OPT}$, where $c_i \in C'_{\ell^*}$ is the closest client to $c$ in $C'_{\ell^*}$, and the last inequality follows since $\lambda^* \leq \text{OPT}$. Finally, note that $S$ may still fail to satisfy the group constraints since $(i)$ $M$ may not match every vertex in $\mathbb{G}$ of $H$, and/or $(ii)$ the requirements are larger than 1. But this can be easily handled by adding arbitrary facilities of each unmatched group to $S$.

Now, to obtain $\lambda^*$, we can do the following. Let $\Gamma$ denote the set of distances from each client in $C'_{\ell^*}$ to $F$. Note that $|\Gamma| \leq |F|\ell \leq nk$. Since $\lambda^*$ is defined to be the largest distance of clients in $C'_{\ell^*}$ to $F$, we have that $\lambda^* \in \Gamma$. Finally, we can do a binary search on the sorted $\Gamma$ to find the smallest distance in $\Gamma$ that returns a feasible matching on $H$.

**Time complexity.** Naively iterating over all values of $\ell \in [k]$ and $\lambda \in \Gamma^\ell$ results in $O(k^3 n^2 + k^3 \sqrt{k}\, n)$ time, instead we adopt an efficient approach by employing binary-search over $\ell \in [k]$ and $\lambda \in \Gamma^\ell$. Although there are at most $\ell \cdot n$ distinct radii in $\Gamma^\ell$, it is not necessary to check if a feasible matching exists for each radius in $\Gamma^\ell$ to find the optimal solution $\lambda^*$. Instead, binary search on sorted $\Gamma^\ell$ is sufficient and can be done in $\log \ell n$ iterations. If a solution exists for some radius $\lambda > \lambda^*$, then we can reduce the radius to check for smaller feasible radii. Conversely, if no feasible solution exists for a radius $\lambda$, we can discard all radii smaller than $\lambda$. Furthermore, by employing binary search on $\ell \in \{1, \dots, k\}$ to find the maximum $\ell^*$ for which a feasible matching on $\{c_1, \dots, c_\ell\}$ exists, reducing the number of iterations to $\log k$. If a matching exists for some $\ell$ and $\lambda$, a matching also exists for the same $\lambda$ and any smaller $\ell$. Conversely, if no matching exists for a given $\ell$ and $\lambda$, then no matching exists for any larger $\ell$ for the fixed vale of $\lambda$. The pseudocode is available in Algorithm 1, where Line 6 is executed for $\log k$ iterations, and Line 8 sorts $\ell n$ elements in $O(\ell n \log \ell n)$, resulting in time complexity of $O(kn \log(kn) \log k)$ for sorting. Further, Line 9 is executed for $\log(n\ell)$ iterations, with the graph $H_\lambda^\ell$ in Line 14 constructed in

$O(n\ell)$ time, and the maximal matching takes $O(k^2 \sqrt{k})$. Thus, the overall time complexity is $O((kn + k^2\sqrt{k}) \log n \log k)$.[7]

**Hardness of approximation.** It is known that [15] for any $\epsilon > 0$ there exists no $3 - \epsilon$ approximation algorithm in polynomial time for $k$-Supplier, assuming P $\neq$ NP. When the number of groups $t$ is equal to 1, Fair-$k$-Sup-$\varnothing$ is equivalent to $k$-Supplier and hence, the hardness of approximation follows. □

Next, we extend our approach to obtain a 3-approximation algorithm for Fair-$k$-Sup, when the facility groups intersect. By combining the methods of Thejaswi et al. [24] and Chen et al. [6], a 5-approximation algorithm can be obtained in time $O(2^{tk}k^2 n^2)$.[8] We improve the approximation ratio to 3 in time $O(2^{tk}tn(kn + k^2\sqrt{k}) \log k \log n)$.

THEOREM 3.2. *There is a 3-approximation for* Fair-$k$-Sup *in time* $O(2^{tk}tn(kn + k^2\sqrt{k}) \log n \log k)$. *Furthermore, assuming* $W[2] \neq$ *FPT, there is no $(3 - \epsilon)$-approximation for* Fair-$k$-Sup *in FPT$(k + t)$ time, for any $\epsilon > 0$.*

**Proof.** The high level idea of our algorithm (see Algorithm 2) is to reduce the given instance $I$ of Fair-$k$-Sup problem to many instances of the same problem but with disjoint groups such that the at least one instance with disjoint groups has same cost as the optimal cost of $I$. Then, we apply Algorithm 1 on each of the reduced instances to find a 3-approximate solution and return the solution $T^*$ corresponding to the instance that has smallest cost. By correctness of the reduction, $T^*$ is a 3-approximate solution to $I$.

In more details, we associate each facility $f \in F$ with a characteristic (bit) vector $\vec{\chi}_f \in \{0,1\}^t$, where the $i$-th index is 1 if $f \in G_i$ and 0 otherwise. For each unique bit vector $\vec{\gamma} \in \{0,1\}^t$, define $Q(\vec{\gamma}) = \{f \in F : \vec{\chi}_f = \vec{\gamma}\}$ as the subset of facilities with characteristic vector $\vec{\gamma}$. The set $\mathcal{P} = \{Q(\vec{\gamma})\}_{\vec{\gamma} \in \{0,1\}^t}$ forms a partition of $F$. Let $F^*$ be an optimal solution to $I$, and let $\{\vec{\gamma}_1^*, \dots, \vec{\gamma}_k^*\} \subseteq \{\vec{\gamma}\}_{\vec{\gamma} \in \{0,1\}^t}$ be the $k$-multiset of bit vectors corresponding to the facilities in $F^*$. Since $F^*$ is feasible, we have $\sum_{i \in [k]} \vec{\gamma}_i^* \geq \vec{\alpha}$ (element-wise). Hence, if we could find $\{\vec{\gamma}_1^*, \dots, \vec{\gamma}_k^*\}$, then we can create an instance $J$ of Fair-$k$-Sup with disjoint groups $\{Q(\vec{\gamma}_1^*), \dots, Q(\vec{\gamma}_k^*)\}$ and run Algorithm 1 on $J$ (see Line 9) to obtain a 3-approximate solution $T$ for $J$. Note that $T$ is also feasible for $I$ and hence a 3-approximate solution for $I$. However, since we do not know $\{\vec{\gamma}_1^*, \dots, \vec{\gamma}_k^*\}$, we enumerate all feasible $k$-multisets of $\mathcal{P}$ (see Line 6), and run Algorithm 1 on the instances corresponding to the enumerated $k$-multisets. Finally, by returning the minimum cost solution (see Line 10) over the all the instances, we make sure that the cost of the returned solution is at most the cost of $T$. □

**Time complexity.** The set $\mathcal{P}$ can be constructed in time $O(2^t n)$ since $|\mathcal{P}| \leq 2^t$. There are $\binom{2^t + k - 1}{k}$ possible $k$-multisets of $\mathcal{P}$, and enumerating them and verifying that they satisfy the range constraints in $\vec{\alpha}$ takes $O(2^{tk}tn)$. For each valid instance, we apply Theorem 3.1 to obtain a 3-approximation, which takes $O((kn +$

---

[7]Precisely, the calculations are as follows $O(kn \log(kn) \log k) + O((kn + k^2\sqrt{k}) \log(kn) \log k) = O((kn + k^2\sqrt{k}) \log n \log k)$.

[8]In Algorithm 2, Line 9 invokes a subroutine for Fair-$k$-Sup-$\varnothing$, which can be substituted with the 5-approximation algorithm from Chen et al. [6], yielding a 5-approximation algorithm with time $O(2^{tk}k^2 n^2)$. Moreover, any improvement in the approximation ratio or runtime for Fair-$k$-Sup-$\varnothing$ would translate to improvements for Fair-$k$-Sup.

$k^2\sqrt{k})\log n \log k)$. Thus, the overall time complexity is $O(2^{tk}tn(kn+ k^2\sqrt{k})\log k \log n)$.[9]

**Hardness of approximation.**[10] It is known that [11] there exists no algorithm that approximates $k$-Supplier to $3-\epsilon$ factor in $\mathrm{FPT}(k)$ time, for any $\epsilon > 0$, assuming $\mathrm{W}[2] \neq \mathrm{FPT}$. When number of groups $t = 1$, Fair-$k$-Sup is equivalent to $k$-Supplier and the hardness of approximation follows by observing that $\mathrm{FPT}(k + t) = \mathrm{FPT}(k)$, for $t = 1$. □

**Solving fair range clustering.** In the literature, the problem variant with restriction on minimum and maximum number of facilities that can be chosen from each group is referred as *fair range clustering*. Our approach can be extended to obtain a 3-approximation for Fair-$k$-Sup with both lower and upper bound requirements, where the number of chosen cluster centers from each group must be within the range specified by lower and upper bound thresholds. Suppose $\vec{\beta} = \{\beta_1, \ldots, \beta_t\}$ represents the upper bound threshold, then, in Line 6 of Algorithm 2, we take into account both $\vec{\alpha}$ and $\vec{\beta}$ for computing the feasibility of the multiset considered in Line 5. Specifically, we change the If condition in Line 6 to the following.

Line 6: **If** $\vec{\alpha} \leq \sum_{i \in [k]} \vec{\gamma}_i \leq \vec{\beta}$ **then**

It is routine to check that the instances corresponding to these feasible multisets are, indeed, instances of Fair-$k$-Sup-$\varnothing$. Therefore, we obtain a 3-approximation for this problem with same runtime.

## 4 Experiments

In this section, we present our experimental setup and datasets used for evaluation, and discuss our findings. The experiments are designed to evaluate the scalability of the proposed algorithms against the baselines, and study the "price of fairness" by comparing the solutions obtained with and without fairness constraints. Additional experimental results are available in Appendix A.

**Experimental setup.** Our implementation is written in python using numpy and scipy packages for data processing. The experiments are executed on a compute server with 2× Intel Xeon E5-2667 v2 processor (2 × 8 cores), 256 GB RAM, and Debian GNU/Linux 12 using a *single core without parallelization*.

**Baselines.** We evaluate our algorithms against the following baselines. First, we consider the 3-approximation algorithm for the $k$-Supplier problem without fairness constraints by Hochbaum and Shmoys [14]. Second, we use the 5-approximation algorithm for Fair-$k$-Sup-$\varnothing$ by Chen et al. [6] for disjoint groups. Lastly, we consider a 5-approximation algorithm for Fair-$k$-Sup with intersecting groups by modifying Algorithm 2. Precisely, instead of invoking Algorithm 1 Line 9, we employ the 5-approximation from [6] as a subroutine.[11]

**Synthetic data.** To evaluate the scalability of the proposed methods and the baselines, we generate synthetic data for various configurations of parameters $n$, $d$, $t$, and $k$, using the random subroutine

---

**Algorithm 1:** 3-approximation algorithm for Fair-$k$-Sup-$\varnothing$

**Input:** $I = (C, F, \mathbb{G} = \{G_1, \ldots, G_t\}, k, \vec{\alpha})$, an instance of Fair-$k$-Sup with disjoint groups

**Output:** $S$, a subset of facilities

1   $S \leftarrow \emptyset$

2   $C' \leftarrow$ choose an arbitrary client $c_1 \in C$

3   **for** $i \in \{2, \ldots, k\}$ **do**   // farthest client recursively

4     $c_i \leftarrow \mathrm{argmax}_{c \in C \setminus C'} d(c, C')$

5     $C' \leftarrow C' \cup \{c_i\}$

6   **Binary-Search** on $\ell \in \{1, \ldots, k\}$ **do**

7     $S^\ell \leftarrow \emptyset$

8     $\Gamma^\ell \leftarrow$ Get-Sorted-Radii$(\{c_1, \ldots, c_\ell\}, F)$   // Sorted distances between $\{c_1, \ldots, c_\ell\}$ and $F$

9     **Binary-Search** on $\lambda \in \Gamma^\ell$ **do**

10       $T^\ell_\lambda \leftarrow \emptyset$

11       $\mathbb{G}' \leftarrow \bigcup_{i \in [t]} \{G_i^1, \ldots, G_i^{\alpha_i}\}$   // $\alpha_i$ vertices for $G_i$

12       $V^\ell_\lambda \leftarrow \{c_1, \ldots, c_\ell\} \cup \mathbb{G}'$

13       For $c_i \in \{c_1, \ldots, c_\ell\}$, add edges $(c_i, G_j^1), \ldots, (c_i, G_j^{\alpha_j})$ to $E^\ell_\lambda$ if there exist $f \in G_j$ s.t. $d(c_i, f) \leq \lambda\}$

14       $H^\ell_\lambda \leftarrow (V^\ell_\lambda, E^\ell_\lambda)$   // Create a Bipartite graph

15       $M^\ell_\lambda \leftarrow$ Max-Matching$(H^\ell_\lambda, \{c_1, \ldots, c_\ell\})$   // Maximum matching in $H^\ell_\lambda$ on $\{c_1, \ldots, c_\ell\}$

16       **if** $M^\ell_\lambda$ *is not a matching on* $\{c_1, \ldots, c_\ell\}$ **then**

17         Continue to Line 9

18       **for** $(c_i, G_j^{j'}) \in M^\ell_\lambda$ **do**

19         $T^\ell_\lambda \leftarrow T^\ell_\lambda \cup \{$arbitrary $f \in G_j$ s.t. $d(c_i, f) \leq \lambda\}$

20       **for** $G_j \in \mathbb{G}$ *such that* $|T^\ell_\lambda \cap G_j| < \alpha_j$ **do**

21         Add $\alpha_j - |T^\ell_\lambda \cap G_j|$ many arbitrary facilities from $G_j$ to $T^\ell_\lambda$   // Make $T^\ell_\lambda$ feasible

22       **if** $\mathrm{cost}(C, T^\ell_\lambda) < \mathrm{cost}(C, S^\ell)$ **then**

23         $S^\ell \leftarrow T^\ell_\lambda$

24     **if** $\mathrm{cost}(C, S^\ell) < \mathrm{cost}(C, S)$ **then**

25       $S \leftarrow S^\ell$

26   **return** $S$

---

from numpy. First, we randomly partition the data points $U$ into clients and facilities. For Fair-$k$-Sup-$\varnothing$ instances, the facilities are randomly partitioned into $t$ disjoint groups. For Fair-$k$-Sup with intersecting groups, first we partition the facilities into disjoint groups, then sample facilities at random and add them to groups to enable intersections.

**Real-world data.** We use the following subset of datasets from the UCI Machine Learning Repository [4]: Heart, Student-mat, Student-perf, National-poll, Bank, Census, Credit-card, and Bank-full. The data is preprocessed by creating one-hop encoding for columns with categorical data and applying min-max normalization to avoid skewing the cluster centers towards features with larger values. For experiments with two disjoint groups ($t = 2$), all

---

[9]More precisely, the time complexity is $O(2^{tk}t(n + (kn + k^2\sqrt{k})\log n \log k) = O(2^{tk}t(kn + k^2\sqrt{k})\log n \log k)$.

[10]Following a similar argument, our result implies that, for any $\epsilon > 0$, there exists no $(2 - \epsilon)$-approximation algorithm in $\mathrm{FPT}(k, t)$-time for Fair-$k$-Center with intersecting groups.

[11]Also, we implemented a brute-force algorithm to find the optimal solution, as expected, it did not scale to large instances.

**Algorithm 2:** 3-approximation algorithm for Fair-$k$-Sup

**Input:** $I = (C, F, \mathbb{G} = \{G_1, \ldots, G_t\}, k, \vec{\alpha})$, an instance of Fair-$k$-Sup

**Output:** $T^*$, a subset of facilities

1 **foreach** $\vec{\gamma} \in \{0, 1\}^t$ **do**
2     $Q(\vec{\gamma}) \leftarrow \{f \in F : \vec{\gamma} = \vec{\chi}_f\}$

3 $\mathcal{P} \leftarrow \{Q(\vec{\gamma}) : \vec{\gamma} \in \{0, 1\}^t\}$
4 $T^* \leftarrow \emptyset$
5 **foreach** *multiset* $\{Q(\vec{\gamma}_1), \cdots, Q(\vec{\gamma}_k)\} \subseteq \mathcal{P}$ *of size* $k$ **do**
6     **if** $\sum_{i \in [k]} \vec{\gamma}_i \geq \vec{\alpha}$, *element-wise* **then**
7        Let $\{Q'(\vec{\gamma}_1), \cdots, Q'(\vec{\gamma}_{k'})\}$ be the set obtained from multiset $\{Q(\vec{\gamma}_1), \cdots, Q(\vec{\gamma}_k)\} \subseteq \mathcal{P}$ after removing duplicates
8        Let $\alpha'_i$ be the number of times $Q(\vec{\gamma}_i)$ appear in the multiset $\{Q(\vec{\gamma}_1), \cdots, Q(\vec{\gamma}_k)\}$
9        Let $T$ be the set returned by Algorithm 1 on $(C, F, \{Q'(\vec{\gamma}_1), \cdots, Q'(\vec{\gamma}_{k'})\}, k, \vec{\alpha}' = (\alpha'_1, \ldots, \alpha'_{k'}))$
10        **if** $cost(C', T) < cost(C', T^*)$ **then**
11           $T^* \leftarrow T$

12 **return** $T^*$

data points are treated as clients, and a subset of suppliers (facilities) is selected based on a protected attribute: age $\leq 50$ in Heart, guardian = 'mother' in Student-mat and Student-perf, race = 'Black' in National-poll and Census, education = 'secondary' in Bank and Bank-full, and married = 'True' in Credit-card. The facilities are partitioned into two groups based on the attribute sex. For experiments with multiple disjoint groups ($t = 5$), all data points are considered as clients and suppliers are selected based on attribute sex, except in Bank and Bank-full, where education = 'secondary' is used. The minority partition is considered as facilities and groups are partitioned based on age, except in National-poll, where race is used to create groups.

**Scalability of Algorithm 1 in synthetic data.** We first evaluate the scalability of algorithms for the Fair-$k$-Sup-$\varnothing$ problem on synthetic data. We compare our 3-approximation algorithm (Algorithm 1) with the 5-approximation algorithm of Chen et al. [6]. The results are illustrated in Figure 1. On the left, we report the mean runtime (solid line) and standard deviation (shaded area) for varying dataset sizes, $n = \{10^4, 2 \cdot 10^4, \ldots, 10^7\}$, where clients and facilities are equally split ($n_c = n_f = \frac{n}{2}$). The number of groups is $t = 5$ and the facilities are randomly partitioned equally among groups, with the number of cluster centers fixed at $k = 10$. All data points have same dimension $d = 5$ and all requirements in $\vec{\alpha}$ are same, which is set to $\frac{k}{t}$.

On the right, we report the mean runtime and standard deviation for different number of cluster centers $k = \{5, 10, \ldots, 50\}$, with fixed dataset size $n = 10\,000$ and an equal split between clients and facilities. The number of groups remains $t = 5$ and the groups consist of equal number of facilities chosen at random. All data points have the same dimension $d = 5$ and all requirements in $\vec{\alpha}$ are same, which is set $\frac{k}{t}$. For each configuration of $n$, $d$, $t$, and $k$, we generate 5 independent instances, and we execute both algorithms

5 times per instance, each time with a different initialization chosen at random. This results in 25 independent executions per configuration, for which we report the mean and standard deviation of running times.

Our implementation of Algorithm 1 achieves significant speedup in running time compared to the algorithm of Chen et al. [6], as expected theoretically, and exhibits small variance across independent executions. Notably, our implementation solves Fair-$k$-Sup-$\varnothing$ instances with ten million data points and ten cluster centers in less than 5 minutes. We terminated some executions due to excessively long running times.

**Scalability and solution quality in real-world data.** In Table 2, we report the running times for the 3-approximation of $k$-Supplier without fairness constraints (Hochbaum and Shmoys [14]), our 3-approximation from Algorithm 1, and the 5-approximation of Chen et al. [6], for Fair-$k$-Sup-$\varnothing$ with disjoint facility groups. The experiments are conducted for $k = 10$ and $k = 20$ with $\vec{\alpha} = \{5, 5\}$ and $\{10, 10\}$, respectively. For each dataset we repeat experiments with 10 random initializations, and report the mean and standard deviation of the running times. Similar experiments are performed by varying $\vec{\alpha}$ to enforce selection from minority groups, but this does not affect significantly the running times. Our implementation of Algorithm 1 solves each real-world instance within one minute for $k = 20$.

Table 3 shows the minimum clustering objective values from 10 independent executions. In our experiments, there is no significant difference in objective values between the fair and unfair versions, or between the 3-approximation and 5-approximation solutions.[12] For experiments with multiple groups we refer the reader to Appendix A Table 4 and Table 5.

**Scalability of Algorithm 2 for intersecting groups.** Last, we evaluate the methods for the Fair-$k$-Sup problem with intersecting facility groups. We compare our method in Algorithm 2 with the 5-approximation algorithm (Thejaswi et al. [24] + Chen et al. [6]). On the left, we display the mean runtime (solid line) and standard deviation (shaded area) for different dataset sizes, $n = \{10^4, 2 \cdot 10^4, \ldots, 10^5\}$, where clients and facilities are split equally ($n_c = n_f = \frac{n}{2}$). The number of groups is $t = 4$, with facilities sampled equally to each group with size $2 \cdot \frac{n_f}{t}$, and $k = 5$ cluster centers, with fairness requirements $\vec{\alpha} = \{2, 2, 2\}$.

On the right, we report the mean runtime and standard deviation for cluster center sizes $k = \{5, 6, 7, 8\}$, with a fixed number of data points $n = 1\,000$, and clients and facilities are split equally. The number of groups remains $t = 4$, with facilities chosen randomly for each group with size $2 \cdot \frac{n_f}{t}$. All data points have the same dimension $d = 5$, and fairness requirements are uniform with $\vec{\alpha} = \frac{k}{t}$. For each configuration, we generate 5 random instances and execute 5 iterations of each algorithm per instance, with a different initialization, resulting in 25 total executions per configuration. We report the mean and standard deviation of the 25 executions.

Our algorithm demonstrates modest scalability in both dataset size $n$ and the number of cluster centers $k$, with low variance across

---

[12]The reported minimum objective values depends on the random initialization. While the theoretical approximation bound holds for each iteration, evaluating the quality of solution and drawing informed insights with limited number of iterations is challenging.

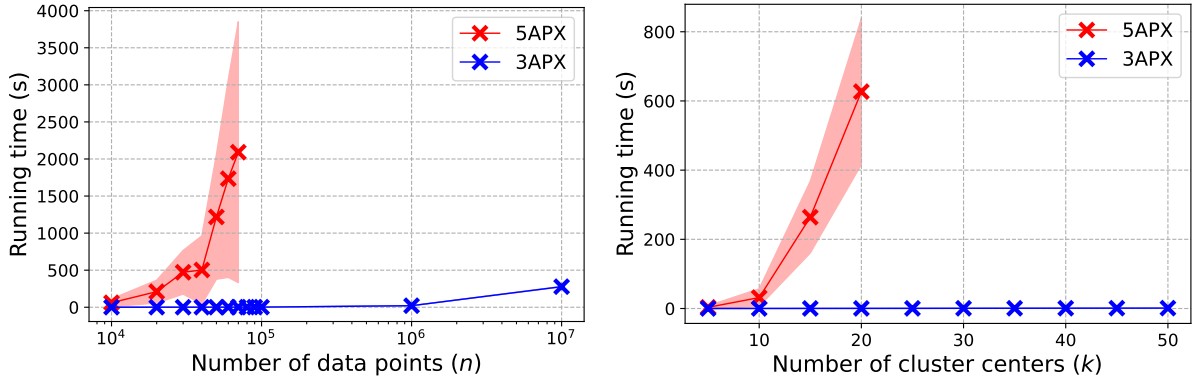

**Figure 1: Scalability of the 3-approximation algorithm (Algorithm 1) and the 5-approximation by Chen et al. [6] for FAIR-$k$-SUP-$\varnothing$ with $t = 5$ disjoint groups and fairness requirements $\vec{\alpha} = \lfloor \frac{k}{t} \rfloor^t$.**

**Table 2: Running time in real-world datasets for FAIR-$k$-SUP-$\varnothing$ with disjoint groups.**

| | | | | | $t = 2$ | $k = 10, \vec{\alpha} = \{5, 5\}$ | | | $k = 20, \vec{\alpha} = \{10, 10\}$ | | |
|---|---|---|---|---|---|---|---|---|---|---|---|
| Dataset | $n$ | $d$ | $n_c$ | $n_f$ | group sizes | 3-apx (unfair) | 3-apx (fair) | 5-apx (fair) | 3-apx (unfair) | 3-apx (fair) | 5-apx (fair) |
| Heart | 299 | 13 | 299 | 74 | (31, 43) | 0.00 ± 0.00 | 0.02 ± 0.00 | 0.06 ± 0.07 | 0.01 ± 0.00 | 0.04 ± 0.00 | 0.10 ± 0.08 |
| Student-mat | 395 | 59 | 395 | 273 | (128, 145) | 0.02 ± 0.00 | 0.08 ± 0.00 | 0.25 ± 0.17 | 0.07 ± 0.00 | 0.18 ± 0.01 | 1.29 ± 1.27 |
| Student-perf | 649 | 59 | 649 | 455 | (182, 273) | 0.03 ± 0.00 | 0.13 ± 0.00 | 0.30 ± 0.22 | 0.08 ± 0.01 | 0.22 ± 0.01 | 1.33 ± 1.74 |
| National-poll | 714 | 50 | 714 | 52 | (23, 29) | 0.01 ± 0.00 | 0.03 ± 0.00 | 0.03 ± 0.01 | 0.04 ± 0.00 | 0.06 ± 0.00 | 0.08 ± 0.02 |
| Bank | 4521 | 53 | 4521 | 2036 | (609, 1427) | 0.15 ± 0.01 | 0.65 ± 0.04 | 11.86 ± 10.00 | 0.66 ± 0.01 | 1.59 ± 0.11 | 18.59 ± 8.32 |
| Credit-card | 30000 | 24 | 30000 | 13659 | (5190, 8469) | 0.51 ± 0.06 | 2.74 ± 0.26 | 38.30 ± 37.37 | 2.20 ± 0.01 | 6.36 ± 0.58 | 223.17 ± 183.65 |
| Bank-full | 45211 | 53 | 45211 | 20387 | (6617, 13770) | 1.70 ± 0.00 | 7.73 ± 0.60 | 245.14 ± 246.94 | 6.63 ± 0.03 | 18.59 ± 1.11 | 1253.49 ± 1279.57 |
| Census | 48842 | 112 | 48842 | 4685 | (2377, 2308) | 3.64 ± 0.01 | 14.02 ± 0.84 | 18.33 ± 27.19 | 14.57 ± 0.06 | 36.09 ± 1.49 | 46.36 ± 42.00 |

**Table 3: Comparison of quality of solutions in real-world datasets for FAIR-$k$-SUP-$\varnothing$ with $t = 2$ disjoint groups.**

| | | | | | $t = 2$ | $k = 10, \vec{\alpha} = \{5, 5\}$ | | | $k = 20, \vec{\alpha} = \{10, 10\}$ | | |
|---|---|---|---|---|---|---|---|---|---|---|---|
| Dataset | $n$ | $d$ | $n_c$ | $n_f$ | group sizes | 3-apx (unfair) | 3-apx (fair) | 5-apx (fair) | 3-apx (unfair) | 3-apx (fair) | 5-apx (fair) |
| Heart | 299 | 13 | 299 | 74 | (31, 43) | 3.63 | **3.58** | 3.72 | **3.00** | 3.00 | 3.46 |
| Student-mat | 395 | 59 | 395 | 273 | (128, 145) | 19.00 | **18.97** | 19.00 | 16.76 | **16.39** | 17.35 |
| Student-perf | 649 | 59 | 649 | 455 | (182, 273) | **18.48** | 19.19 | 19.54 | **17.23** | 17.31 | 18.68 |
| National-poll | 714 | 50 | 714 | 52 | (23, 50) | **14.50** | 14.50 | 16.00 | **14.00** | 14.00 | 14.50 |
| Bank | 4521 | 53 | 4521 | 2036 | (609, 1427) | 12.69 | **12.38** | 12.85 | 11.05 | **10.96** | 12.35 |
| Credit-card | 30000 | 24 | 30000 | 13659 | (5190, 8469) | 6.44 | **6.28** | 6.46 | 5.92 | 5.97 | **5.92** |
| Bank-full | 45211 | 53 | 45211 | 20387 | (6617, 13770) | **12.98** | 12.99 | 13.06 | 11.74 | **11.49** | 12.74 |
| Census | 48842 | 112 | 48842 | 4685 | (2377, 2308) | 15.53 | **15.09** | 15.13 | **13.71** | 14.17 | 14.62 |

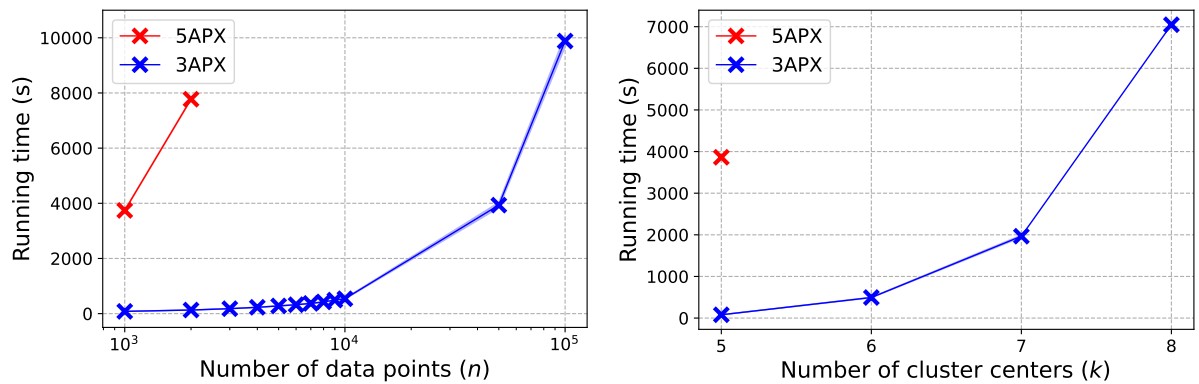

**Figure 2: Scalability of the 3-approximation algorithm (Algorithm 2) and the 5-approximation for FAIR-$k$-SUP with $t = 4$ intersecting groups, where the requirements are uniform across groups with $\vec{\alpha} = [2 \cdot \frac{k}{t}]^t$.**

independent runs. Given the computational complexity of the problem (NP-hard and W[2]-hard with respect to $k$), the modest scalability is expected. Furthermore, our algorithm is significantly more efficient than the baseline, which terminates only for dataset sizes of up to $n = 2 \cdot 10^4$ (left), and only for up to $k = 5$ centers (right).

## 5 Related work

Our work builds upon the existing work on data clustering and algorithmic fairness.

The literature on (fair) clustering is extensive, so we focus on the most relevant research for our work. For a review of clustering, see Jain et al. [17], and for fair clustering, see Chhabra et al. [7]. $k$-Center and $k$-Supplier have been widely studied and numerous algorithmic results are known [10, 11, 13, 15, 23]. Both problems are known to be NP-hard [26], and polynomial-time approximation algorithms have been developed [10, 13, 23]. For $k$-Center and $k$-Supplier, polynomial-time approximation algorithms with factors 2 and 3 are known [14, Theorem 5] [10, Theorem 2.2]. Assuming $P \neq NP$, $k$-Center and $k$-Supplier cannot be approximated within factors of $2 - \epsilon$ and $3 - \epsilon$, respectively, for any $\epsilon > 0$ [14, Theorem 6] [10, Thoerem 4.3]. In the context of fixed-parameter tractability (FPT), $k$-Center and $k$-Supplier are at least W[2]-hard with respect to $k$, meaning that no algorithm with running time $f(k) \cdot \text{poly}(n, k)$ can solve them optimally; this is implicit in a reduction presented by Hochbaum and Shmoys [14]. Assuming FPT $\neq$ W[2], $k$-Center and $k$-Supplier cannot be approximated within factors of $2 - \epsilon$ and $3 - \epsilon$, for any $\epsilon > 0$, in $f(k) \cdot \text{poly}(n, k)$ time, even when $f(k)$ is an exponential function [11, Theorem 2, Theorem 3].

Fairness in clustering has recently attracted significant attention as a means to reduce algorithmic bias in automated decision-making for unsupervised machine-learning tasks. Various fairness notions have been explored, leading to many algorithmic results [1, 2, 5, 8, 9, 12, 22]. Our focus is on cluster center fairness, where data points are associated with demographic attributes forming groups, and fairness is applied to the selection of cluster centers while optimizing different clustering objectives such as $k$-median, $k$-means, $k$-center, and $k$-supplier. Several problem formulations study different types of constraints on the number of cluster centers chosen from each group: exact requirement [18, 21], lower bound [24, 25], upper bound [5, 12, 22], and combined upper and lower bound [3, 16].

In the context of fair data summarization, much of the existing literature focuses on the case where demographic groups are disjoint. Kleindessner et al. [21] introduced the fair $k$-center problem with disjoint groups, [13] where a specified (exact) number of cluster centers must be chosen from each group and the groups were explicitly disjoint. They presented a $3 \cdot 2^{t-1}$ approximation algorithm in $O(nkt^2 + kt^4)$ time, which was later improved to factor 3 in time $O(nk + n\sqrt{k} \log k)$ by Jones et al. [18]. Angelidakis et al. [3] considered a variant of Fair-$k$-Center-$\varnothing$ with both lower and upper bounds on number of cluster centers from each group and presented a 15-approximation in time $O(nk^2 + k^5)$. Chen et al. [5] studied

the matroid $k$-center-$\varnothing$ problem that generalizes Fair-$k$-Center-$\varnothing$, where the chosen cluster centers must form an independent set in a given matroid and presented a 3-approximation algorithm that runs in poly$(n)$ time.[14]

Chen et al. [6] studied the Fair-$k$-Sup-$\varnothing$ problem when the cluster centers must be chosen from a subset of data points called facilities or suppliers and present a 5-approximation in time $O(kn^2 + k^2\sqrt{k})$. Thejaswi et al. [24] studied Fair-k-Median and Fair-$k$-Means with intersecting facility groups and showed that the problem is inapproximable to any multiplicative factor in polynomial time.[15] On the other hand, it is worth noting that their techniques can be extended to obtain a fixed parameter tractable 5-approximation algorithm with a runtime of $O(2^{tk}k^2n^2)$ by leveraging the polynomial-time 5-approximation algorithm of Chen et al. [6] for Fair-$k$-Sup-$\varnothing$ as a subroutine.

## 6 Conclusions, limitations and open problems

**Conclusions.** In this paper, we provide a comprehensive analysis of the computational complexity for Fair-$k$-Sup in terms of its approximability. Specifically, for the case with disjoint groups, we present a *near-linear* time 3-approximation algorithm. For the more general case where the groups may intersect, we present a fixed-parameter tractable 3-approximation algorithm with runtime FPT$(k + t)$. We also show that the approximation factors can not be improved for both the problems, assuming standard complexity conjectures. Additionally, we rigorously evaluate the performance of our algorithms through extensive experiments on both real-world and synthetic datasets. Notably, for the intersecting case, our algorithm is the first with theoretical guarantees (on the approximation factor) while scaling efficiently to instances of modest size, where the earlier works with theoretical guarantees struggled to scale in practice.

**Limitations.** Although our algorithm for intersecting groups scales to modest-sized instances, designing algorithms that can handle web-scale datasets with millions to billions of points remains an open challenge. Limited experiments on real-world data are insufficient to assess the cost of enforcing fairness constraints on solution quality (*i.e.*, clustering objective), as it depends on the specific instance as well as the use case. Drawing a more informed conclusion would require a detailed case study with domain-specific insights, which is beyond the scope of this work. Our focus is to present approximation algorithms with theoretical guarantees for Fair-$k$-Sup that also scale effectively to real-world data.

**Open problems.** For Fair-$k$-Center, while the lower-bound of FPT$(k, t)$-time approximation is 2, but our results imply a 3-approximation algorithm in FPT$(k, t)$ time. An interesting open problem is to either improve the approximation factor for Fair-$k$-Center or to prove that no such improvement is possible. Similarly, for Fair-$k$-Sup (and Fair-$k$-Sup-$\varnothing$), improving the approximation factor for special metric spaces, such as Euclidean spaces, is also a promising direction. Finally, it remains an open question whether or not a linear time algorithm can be designed for Fair-$k$-Sup-$\varnothing$.

---

[13]In the literature, the problem is referred to as fair $k$-center (Fair-$k$-Center), even when only disjoint facility groups are considered. In this work, we distinguish between the two cases, referring to the case with disjoint groups as Fair-$k$-Center-$\varnothing$ and intersecting groups as Fair-$k$-Center.

[14]Though the exact running time is not detailed in Chen et al. [5], it is estimated to be $\Omega(n^2 \log n)$ by Kleindessner et al. [21].

[15]In fact, their complexity results hold for any clustering objective.

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

# A  Additional experimental results

**Scalability of Algorithm 1 on synthetic data.** In Figure 3, we report the running time of Algorithm 1 as the number of clients and facilities increases, while all other parameters remain constant. On the left, we report the mean and standard deviation of 25 independent executions (5 instances and 5 executions per instance) for each $n_c = \{10^2, \ldots, 10^6\}$ with $n = 10^6$, $d = 5$, $k = 10$, $t = 5$, and $\vec{\alpha} = \{2, 2, 2, 2, 2\}$. On the right, we vary $n_f = \{10^2, \ldots, 10^6\}$ while keeping $n$, $d$, $t$, $k$, and $\vec{\alpha}$ fixed. In both cases, we observe increase in running times as $n_c$ and $n_f$ grow.

**Experiments on real-world data.** In Table 4, we report the mean and standard deviation of running times for 10 independent executions for each dataset by considering $t = 5$ disjoint groups for $k = 10$ with requirement vector $\vec{\alpha} = \{4, 3, 1, 1, 1\}$ and $\vec{\alpha} = \{6, 1, 1, 1, 1\}$. We observed difference in running times when we constrained the requirement vector to choose more facilities from minority group.

In Table 5, we report the minimum value of the clustering objective from 10 independent executions for each dataset by considering $t = 5$ disjoint groups for $k = 10$ with requirement vector $\vec{\alpha} = \{4, 3, 1, 1, 1\}$ and $\vec{\alpha} = \{6, 1, 1, 1, 1\}$. Again, we do not observe significant difference in objective values between the fair and unfair versions, nor between the 3 and 5 approximations with fairness constraints.

# B  Proof of Theorem 3.1

Let OPT be the optimal cost of the input instance $I$ to Algorithm 1. Fix an optimal clustering $C^* = \{C_1^*, \ldots, C_k^*\}$ to $I$ corresponding to the solution $F^* = \{f_1^*, \ldots, f_k^*\}$. Consider $C' = (c_1', \ldots, c_k')$ constructed at the end of the for loop in line line 3. We claim that $d(c, C') \leq 2 \cdot \text{OPT}$, for every $c \in C$. Suppose $c_i' \in C_i^*$ for all $c_i' \in C'$. In this case, consider $c \in C$ such that $c \in C_i^*$, and hence $d(c, F^*) \leq d(c, f_i^*) \leq \text{OPT}$. Therefore, $d(c, C') \leq d(c, c_i') \leq d(c, f_i^*) + d(f_i^*, c_i') \leq 2 \cdot \text{OPT}$, by triangle inequality, and the fact that $c_i', c \in C_i^*$. Now, suppose that there is $C_i^* \in C^*$ such that $C_i^* \cap C' = \emptyset$, this means there exist $C_j^* \in C^*$ such that $|C_j^* \cap C'| \geq 2$ since both $C'$ and $C^*$ are of size $k$. Let $c_a', c_b' \in C' \cap C_j^*$ such that $c_a'$ was added to $C'$ before $c_b'$. Furthermore, let $C'' = (c_1', \ldots, c_{b-1}')$ be the set maintained by the algorithm just before adding $c_b'$ to $C'$. Then, note that $d(c_b', C'') \leq d(c_b', c_a') \leq 2 \cdot \text{OPT}$. Since the algorithm selected $c_b'$ to be the furthest point from $C''$, it holds that, for any $c \in C$, we have $d(c, C') \leq d(c, C'') \leq d(c_b', C'') \leq 2 \cdot \text{OPT}$, as required.

The next phase of the algorithm obtains a feasible solution from $C'$. Towards this, the algorithm identifies (by binary search) the smallest index $\ell^*$ such that each point in $C'_{\ell^*-1} := (c_1', \ldots, c_{\ell^*-1}')$ belongs to a unique cluster in $C^*$, but $C'_{\ell^*} := (c_1', \ldots, c_{\ell^*}')$ does not have this property.[16] Next, the algorithm (again using binary search) finds $\lambda^*$, which is defined as the maximum distance between any point in $C'_{\ell^*-1}$ and $F^*$. With $\ell^*$ and $\lambda^*$ in hand, the algorithm constructs a bipartite graph $H_{\lambda^*}^{\ell^*} = (V_{\lambda^*}^{\ell^*}, E_{\lambda^*}^{\ell^*})$ a follows. The left partition of $V_{\lambda^*}^{\ell^*}$ contains a vertex for every point in $C'_{\ell^*-1}$, while the right partition contains $\alpha_j$ vertices $\{G_j^1, \ldots, G_j^{\alpha_j}\}$ for every $G_j \in \mathbb{G}$. For each $c_i' \in C'_{\ell^*-1}$ and $G_j \in \mathbb{G}$, add edges between the vertex $c_i'$ and all vertices $\{G_j^1, \ldots, G_j^{\alpha_j}\}$ if there exists a facility in $G_j$ at

---

[16]We let $\ell^* = k + 1$, for the corner case.

**Table 4: Comparison of running times for real-world datasets for Fair-$k$-Sup-$\varnothing$ with $t = 5$ disjoint groups and different requirement vectors.**

| Dataset | $n$ | $d$ | $n_c$ | $n_f$ | $t = 5$ group sizes | $k = 10, \vec{\alpha} = \{4, 3, 1, 1, 1\}$ 3-apx unfair | 3-apx fair | 5-apx fair | $k = 10, \vec{\alpha} = \{6, 1, 1, 1, 1\}$ 3-apx unfair | 3-apx fair | 5-apx fair |
|---|---|---|---|---|---|---|---|---|---|---|---|
| Heart | 299 | 13 | 299 | 194 | (27, 56, 58, 34, 19) | 0.00 ± 0.00 | 0.03 ± 0.01 | 2.22 ± 1.53 | 0.00 ± 0.00 | 0.03 ± 0.00 | 2.61 ± 1.25 |
| Student-mat | 395 | 59 | 395 | 208 | (15, 38, 43, 54, 58) | 0.02 ± 0.00 | 0.10 ± 0.00 | 3.51 ± 1.57 | 0.02 ± 0.00 | 0.10 ± 0.00 | 3.80 ± 1.35 |
| Student-perf | 649 | 59 | 649 | 383 | (24, 57, 84, 105, 113) | 0.04 ± 0.00 | 0.16 ± 0.01 | 7.02 ± 3.13 | 0.04 ± 0.00 | 0.17 ± 0.01 | 7.75 ± 3.13 |
| National-poll | 714 | 50 | 714 | 393 | (8, 12, 29, 29, 315) | 0.01 ± 0.00 | 0.04 ± 0.00 | 0.15 ± 0.02 | 0.02 ± 0.00 | 0.04 ± 0.01 | 0.17 ± 0.02 |
| Bank | 4521 | 53 | 4521 | 2306 | (64, 302, 383, 609, 948) | 0.17 ± 0.01 | 0.69 ± 0.05 | 76.03 ± 31.78 | 0.17 ± 0.00 | 0.70 ± 0.05 | 75.82 ± 31.05 |
| Census | 48842 | 112 | 48842 | 16192 | (1276, 1873, 3188, 3853, 6002) | 3.37 ± 0.19 | 15.11 ± 1.08 | 3303.58 ± 2173.41 | 3.29 ± 0.21 | 14.75 ± 1.09 | 4981.32 ± 2747.76 |
| Credit-card | 30000 | 24 | 30000 | 11888 | (179, 1092, 2771, 3281, 4565) | 0.39 ± 0.03 | 1.71 ± 0.13 | 486.48 ± 273.89 | 0.31 ± 0.06 | 1.35 ± 0.17 | 676.08 ± 304.78 |
| Bank-full | 45211 | 53 | 45211 | 23202 | (672, 3207, 3851, 6011, 9461) | 1.41 ± 0.01 | 6.73 ± 0.40 | 4542.11 ± 1604.71 | 1.44 ± 0.05 | 6.75 ± 0.81 | 4132.72 ± 2011.38 |

**Table 5: Comparison of quality of solutions in real-world datasets for Fair-$k$-Sup-$\varnothing$ with $t = 5$ disjoint groups.**

| Dataset | $n$ | $d$ | $n_c$ | $n_f$ | $t = 5$ group sizes | $k = 10, \vec{\alpha} = \{4, 3, 1, 1, 1\}$ 3-apx (unfair) | 3-apx (fair) | 5-apx (fair) | $k = 10, \vec{\alpha} = \{6, 1, 1, 1, 1\}$ 3-apx (unfair) | 3-apx (fair) | 5-apx (fair) |
|---|---|---|---|---|---|---|---|---|---|---|---|
| Heart | 299 | 13 | 299 | 194 | (27, 56, 58, 34, 19) | **3.79** | 3.90 | 4.10 | **3.79** | 4.12 | 3.96 |
| Student-mat | 395 | 59 | 395 | 208 | (15, 38, 43, 54, 58) | **19.73** | 20.14 | 20.07 | **19.73** | 20.34 | 20.07 |
| Student-perf | 649 | 59 | 649 | 383 | (24, 57, 84, 105, 113) | 19.69 | **19.50** | 19.55 | 19.69 | **19.49** | 19.71 |
| National-poll | 714 | 50 | 714 | 393 | (8, 12, 29, 29, 315) | **14.50** | 14.50 | 16.00 | **14.50** | 15.00 | 16.00 |
| Bank | 4521 | 53 | 4521 | 2306 | (64, 302, 383, 609, 948) | **12.27** | 12.59 | 12.85 | **12.27** | 12.65 | 12.75 |
| Census | 48842 | 112 | 48842 | 16192 | (1276, 1873, 3188, 3853, 6002) | 16.41 | **15.60** | 17.02 | 16.41 | **16.05** | 17.02 |
| Credit-card | 30000 | 24 | 30000 | 11888 | (179, 1092, 2771, 3281, 4565) | 6.94 | 6.94 | **6.87** | 6.94 | **6.84** | 7.03 |
| Bank-full | 45211 | 53 | 45211 | 23202 | (672, 3207, 3851, 6011, 9461) | **12.87** | 12.97 | 13.05 | 12.87 | **12.67** | 13.05 |

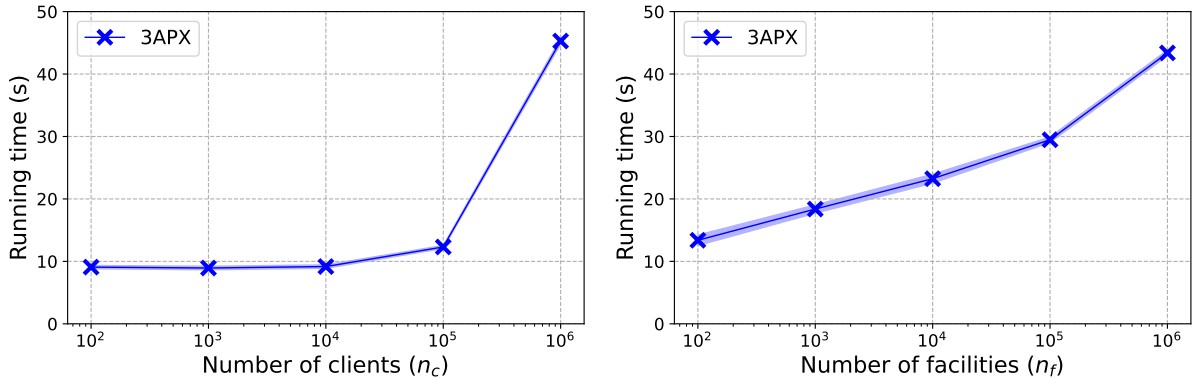

**Figure 3: Scalability of the $3$-approximation algorithm (Algorithm 1) with respect to number of clients $n_c$ and number of facilities $n_f$ for Fair-$k$-Sup-$\varnothing$ with $t = 5$ disjoint groups and fairness requirements $\vec{\alpha} = [\frac{k}{t}]^t$.**

a distance $\lambda^*$ from $c'_i$ (see lines 12-14). The following is the key lemma that is crucial for the correctness of our algorithm.

LEMMA B.1. *There is a matching in $H^{\ell^*}_{\lambda^*}$ on its left partition.*

PROOF. For ease of presentation, suppose the left partition of $H^{\ell^*}_{\lambda^*}$ is denoted as $C'_{\ell^*-1} = (c'_1, \ldots, c'_{\ell^*-1})$. Then note that $|C'_{\ell^*-1}| \leq k = |\mathbb{G}'|$, where $\mathbb{G}'$ (line 12) is the right partition of $H^{\ell^*}_{\lambda^*}$. Let $F^*_j = \{f^1_j, \ldots, f^{\alpha_j}_j\}$ be the facilities in $F^* \cap G_j$, for $G_j \in \mathbb{G}$. Now, consider point $c'_i \in C'_{\ell^*-1}$ and let $f^{j'}_j \in F^*_j$ be the optimal facility in $F^*$ that is closest to $c'_i$. Then, note that $d(c'_i, f^{j'}_j) \leq \lambda^*$, by definition of $\lambda^*$. Hence, there is an edge between vertex $c'_i$ and $G^{j'}_j$ in $H^{\ell^*}_{\lambda^*}$. Since each

$c'_i \in C'_{\ell^*-1}$ belongs to different cluster in $C^*$ and $|C'_{\ell^*-1}| \leq |\mathbb{G}'|$, we have that there is a matching in $H^{\ell^*}_{\lambda^*}$ on $C'_{\ell^*-1}$, as desired. □

Let $M$ be a matching in $H^{\ell^*}_{\lambda^*}$ on its left partition. Let $T^{\ell^*}_{\lambda^*} \subseteq F$ obtained (at the end of the for loop at line 18) by taking an arbitrary facility from $G_j$, for every $(c'_i, G^{j'}_j) \in M$. Then, note that $d(c'_i, T^{\ell^*}_{\lambda^*}) \leq \lambda^* \leq \text{OPT}$, for every $c'_i \in C'_{\ell^*-1}$. Therefore, $d(c, T^{\ell^*}_{\lambda^*}) \leq 3 \cdot \text{OPT}$, as required. Finally, we add as many arbitrary facilities from each $G_j \in \mathbb{G}$ to $T^{\ell^*}_{\lambda^*}$ (line 21) so that $|T^{\ell^*}_{\lambda^*} \cap G_j| = \alpha_j$. This completes the proof.

