# OpenReview forum: "Fair Clustering for Data Summarization: Improved Approximation Algorithms and Complexity Insights"
_ACM.org/TheWebConf/2025/Conference — WWW 2025 Poster_

### Official Review · Reviewer_HBDG · 2024-11-25

**Novelty:** 2
**Technical Quality:** 2

**Review:**

In the manuscript titled “Fair Clustering for Data Summarization: Improved Approximation Algorithms and Complexity Insights”, the authors analyze a 3-approximation algorithm for the Fair-k-Supplier problem, providing insights into scalability and fairness trade-offs. However, the heavy reliance on synthetic data, limited real-world datasets, lack of detailed fairness impact analysis, and insufficient complexity evaluation across parameters significantly weaken the study's applicability and conclusions. Therefore, I believe that this article should not be accepted.

**Questions:**

The following are my reasons:
1. In the second phase of the algorithm, large-scale graphs (e.g., the bipartite graph HHH) are created, which might consume a significant amount of memory. As nnn and kkk increase, will the memory requirements also increase substantially, potentially leading to memory overflow or a decline in system performance?
2. For multi-group problems , the algorithm needs to enumerate all possible kkk-multisets, resulting in a high computational cost. Could this limit the algorithm’s applicability to large-scale problems?
3. The algorithm is designed with assumptions about certain input data structures, such as grouping operations. Could this lead to efficiency issues or failures when applied to data that does not meet these assumptions?
4. The selected baseline algorithms, such as the 3-approximation and 5-approximation, have high complexity and computational cost. If the goal is to demonstrate the advantages of the new algorithm, should more efficient and widely-used baselines be included, or should a wider variety of baselines be considered to verify the proposed method's general applicability?
5. Although the computational complexity of the problem is mentioned, there is no further discussion on the actual time and space complexity of the algorithm at different data scales. Should there be a more detailed quantitative analysis of the algorithm's complexity, particularly regarding how various parameter configurations impact its performance?
6. The experiments use a large amount of synthetic data to evaluate the algorithm's scalability, while the real datasets are smaller and limited in number. Since synthetic data may not fully capture the complexity and characteristics of real data, especially under fairness constraints, could this lead to results that are not directly applicable to real-world scenarios? Would testing the algorithm on more large-scale real datasets better validate its scalability?
7. The experiments mention analyzing the "price of fairness," but the charts do not clearly show the specific impact of fairness constraints on solution quality and algorithm running time. While Table 3 provides a comparison between fair and unfair constraints, it lacks an in-depth analysis of how fairness constraints affect runtime or solution quality. Could a more detailed analysis of the price of fairness strengthen the persuasiveness of the experimental results in this area?

**Reviewer Confidence:**

4: The reviewer is certain that the evaluation is correct and very familiar with the relevant literature

**Scope:**

3: The work is somewhat relevant to the Web and to the track, and is of narrow interest to a sub-community

---

### Official Review · Reviewer_XzQz · 2024-11-29

**Novelty:** 6
**Technical Quality:** 6

**Review:**

This work studies fair clustering for data summarization, where data includes intersectional groups or disjoint demographic groups. The main challenge arises from the increasing computational complexity of fair clustering with intersectional groups. To address this challenge, this work presents a 3-approximation algorithm where the exponential runtime depends on the number of groups and centers, improving the previously known factor of 5. Both simulation and empirical experiments demonstrate the scalability of the algorithms.

Strengths:
- The work proposed the first algorithm that provides a theoretical guarantee while improving the efficiency for the intersectional group case.
- The studied problem is important and less researched. The theoretical analyses are solid and provide important insights, especially when there are intersectional groups.
- Experiments are comprehensive, including large-scale synthetic data and real-world data.
- Major limitations are discussed, which provide potential directions for others to explore.
- Paper is well-written.

Weaknesses:
- It would be more convincing to test on large-scale real-world data, which is understandably hard to access. Would it be possible to generate semi-synthetic data based on the small real-world datasets?
- Missing reference

Gohar, U., & Cheng, L. (2023, August). A survey on intersectional fairness in machine learning: notions, mitigation, and challenges. In Proceedings of the Thirty-Second International Joint Conference on Artificial Intelligence (pp. 6619-6627).

**Questions:**

See above.

**Reviewer Confidence:**

2: The reviewer is willing to defend the evaluation, but it is likely that the reviewer did not understand parts of the paper

**Scope:**

4: The work is relevant to the Web and to the track, and is of broad interest to the community

---

### Official Review · Reviewer_XbeV · 2024-11-30

**Novelty:** 5
**Technical Quality:** 5

**Review:**

This paper considers the problem of fair data summarization modeled as a fair-k supplier problem, where data is categorized into several groups and the cluster centers must be chosen from a specific subset of data points while ensuring fair representation across groups and minimizing the maximum distance from the data points to their closest cluster center. The authors also propose to consider both disjoint and overlapping groups and provide approximation algorithms for both variants, improving on the existing algorithms. The proposed method is efficient and scales to large datasets.

I do not have any concern about the method, but I fail to see how this is related to the responsible Web track. The datasets or the task considered in the paper are also not quite related to Web data. While the authors consider an anecdotal example to better motivate the problem, I fail to see the connection.

The literature in fair clustering is also quite extensive and it is important to contextualize the paper's contribution with respect to the existing literature. How is the problem considered here different from the ones already considered in the existing fair clustering literature? This will perhaps also help in motivating the problem.

I was also a bit unsure about the experiments with the real-world datasets. Firstly, how do you measure the quality of the cluster in terms of both utility and fairness? Secondly, what exactly is your fairness metric? I appreciate your complexity results but as someone with a more empirical background, I would be more interested in the practicalities of the proposed algorithms. For the real-world datasets, what would be a perfectly fair clustering? You can consider one of the datasets as an example and explain it.

**Questions:**

You provided results on how the algorithms scale with the number of data points. But what about the number of groups? Given you are considering intersectional groups, this might also expand really quickly depending on the number of attributes. For example, if I consider age with 10 possible groups and race with say 4 groups, the number of intersectional groups would be 40.

Also, what happens if the population of these groups are skewed, which is often encountered in the real world?

**Reviewer Confidence:**

3: The reviewer is confident but not certain that the evaluation is correct

**Scope:**

2: The connection to the Web is incidental, e.g., use of Web data or API

---

### Official Review · Reviewer_dQSb · 2024-11-30

**Novelty:** 3
**Technical Quality:** 5

**Review:**

Overall
The authors introduce a new, more efficient algorithm (with theoretical guarantees) for performing fair clusterings.  I appreciate the topic and the authors’ efforts to improve the existing algorithmic state of the art on the topic.

I think my biggest challenges with this paper are two-fold:

1. Motivation of problem and contextualization of results — I had hard time throughout the paper appreciating why this topic deserves study.  There are already fair clustering algorithms, as highlighted by the authors.  They may be less efficient / scalable than what the authors introduce, but why does that matter?  How is web research currently being impeded by this limitation?  What will we be able to do now (with the contributions the authors make), in a practical sense, that we couldn’t do before?  Spending more time on these topics and even giving examples / orienting their empirical evaluations around them could help.

2. Clarity of results — I found the tables and figures hard to interpret.  Which algorithms are the authors’?  Which are baselines?  What’s the magnitude / scale of differences in quality and outputs (especially in relation to what’s presented in the tables — the different results don’t seem that different)?  Figure 2 (the line plots) help … but why are only two of the three algorithms depicted there?  Also, why are comparisons being made across fair and unfair algorithms with different approximation multipliers?  It feels like an apples to oranges comparison at the moment.  Is it possible to run the exact same approximation multiplier, with fairness, using different algorithms to really accentuate the contributions of the authors’ new and improved methods?

Quality
I appreciate that the paper is well-written.  Most of my concerns re: quality are captured above and in the sections below.

Clarity
I found the technical details a bit hard to follow at times (that may be because of my own familiarity with the topic, though), and sometimes I felt like there was too much focus on the technical details at the expense of motivating the topic/problem at hand and why it matters.  I also found some challenges in interpreting the results — for example, the tables/figures don’t make it immediately clear what the authors’ algorithm is vs. what their comparison baselines are (therefore, the improvements are hard to glean from a quick glance); what units performance is being measured in (seconds I believe?); etc.

Originality
The paper appears to build on an existing algorithm for fair clustering.  In that sense, it isn’t new, but it does seem to introduce new methods for performing this clustering more efficiently.

Significance
The study feels like a useful contribution to the algorithms literature, but I am not able to tell if the performance gains it introduces would be useful / relevant in a real-world application setting.  The algorithm might be faster/more efficient, but does it matter for particular applications?  Can we just use the older (slower) algorithms and essentially achieve the same downstream outcomes?

Detailed comments
* Abstract — I think this can be shortened and tightened a bit to focus more on the problem/gap being addressed.  For example, what is meant by “fair”?  Why is this needed in the context of clustering?  What’s the problem this paper is trying to solve and why does it matter?  Ideally, the abstract speaks to these points.
* Introduction — I think the description of the algorithm should be reserved for later (I understand that the description in the intro is high-level, but it still seems to break the flow of “what we did and why it matters” that I think a sound intro would achieve)
* Experiments — It’s unclear in the tables which algorithm is the authors’; ideally this is highlighted more clearly to make the contrast with baselines more salient.  Also, it’s unclear from these tables / Figure 2 alone what the units of time are (minutes? Seconds? etc).  I see in the Figure it might be seconds … I am assuming the same for the tables?
* Discussion — I would love to see a discussion section that reflects on the results and potential real-world implications of these findings.  How stark are the performance gains in terms of real-world applications?  What can we do now with these new algorithms in hand that we couldn’t do before, or do as well before, they existed?  It’s hard to contextualize the contribution without this discussion.

**Questions:**

Questions are included in the review above.

**Reviewer Confidence:**

2: The reviewer is willing to defend the evaluation, but it is likely that the reviewer did not understand parts of the paper

**Scope:**

3: The work is somewhat relevant to the Web and to the track, and is of narrow interest to a sub-community

---

### Official Review · Reviewer_dYbB · 2024-12-02

**Novelty:** 5
**Technical Quality:** 5

**Review:**

Strong points:
- The paper proposed 3-approximation algorithms to solve the fair k-supplier problem. The algorithms are evaluated on 1 synthetic dataset and 8 real-world datasets. The experimental results show that the proposed method outperform the competitors.
- The proposed method is theoretically guaranteed.
Weak points:
- The paper mentioned the work [21] in the related work, but it is not considered as a competitor. Besides, the paper should compare the proposed method with the traditional clustering models, such as k-center.
- The experiments are performed on non-Web data.
- In the experiments, the authors do not explain how/why the parameter are chosen, such as the range of k.

**Questions:**

Please refer to the weak points.

**Reviewer Confidence:**

3: The reviewer is confident but not certain that the evaluation is correct

**Scope:**

2: The connection to the Web is incidental, e.g., use of Web data or API